# Extracellular Vesicle Characteristics in Local Fluid and Plasma Measured by Nanoparticle Tracking Analysis Can Help Differentiate High-Grade Serous Carcinoma from Benign Ovarian Pathology

**DOI:** 10.3390/diagnostics14192235

**Published:** 2024-10-07

**Authors:** Maruša Herzog, Ivan Verdenik, Katarina Černe, Borut Kobal

**Affiliations:** 1Division of Gynecology and Obstetrics, University Medical Centre Ljubljana, SI-1000 Ljubljana, Slovenia; marusa.herzog@kclj.si (M.H.); ivan.verdenik@guest.arnes.si (I.V.); 2Institute of Pharmacology and Experimental Toxicology, Faculty of Medicine, University of Ljubljana, SI-1000 Ljubljana, Slovenia; 3Department of Gynecology and Obstetrics, Faculty of Medicine, University of Ljubljana, SI-1000 Ljubljana, Slovenia

**Keywords:** ovarian cancer, high-grade serous carcinoma, extracellular vesicles, nanoparticle-tracking analysis, diagnostic biomarkers, liquid biopsy

## Abstract

**Background:** High-grade serous carcinoma (HGSC) is the most lethal of gynecological cancers in developed countries. It usually presents late with non-specific symptoms and most cases are diagnosed at an advanced stage, with 5-year overall survival being around 40%. Biomarkers for screening and early diagnosis of this aggressive disease are, thus, a research priority. Extracellular vesicles (EVs) that reflect the cell of origin and that can be isolated from local fluid and plasma by minimally invasive liquid biopsy are such promising biomarkers. Besides EV concentration and molecular profile, which have been the main focus of research for many years, recent studies have also called attention to EV size distribution. The aim of our study was to evaluate the potential of EV concentration and size distribution in local fluid and plasma as diagnostic biomarkers for HGSC. **Methods:** Paired pretreatment ascites and plasma samples from 37 patients with advanced HGSC and paired pretreatment free peritoneal fluid (FPF) and plasma samples from 40 controls with benign ovarian pathology (BOP) were analyzed using nanoparticle tracking analysis (NTA). **Results:** We observed a significant difference in EV concentration in local fluid, but not in plasma, between HGSC patients and the control group. We also found a significant difference in EV size distribution in both local fluid and plasma between HGSC patients and the control group. The receiver operating characteristics (ROC) curve analysis of EV characteristics showed excellent diagnostic performance for the mode, D10, and D50 in local fluid and acceptable diagnostic performance for EV concentration and mean EV size in local fluid, as well as for the mode and D10 value in plasma. **Conclusions:** The results of our study show that EV concentration in local fluid and more importantly EV size distribution in both local fluid and plasma are significantly changed in the presence of HGSC. Future research of size-dependent molecular profiling of EVs could help identify novel diagnostic biomarkers for HGSC.

## 1. Introduction

Ovarian cancer is the most lethal of gynecological cancers in developed countries [1]. The most common type, which also has the worst prognosis, is high-grade serous carcinoma (HGSC). Due to late onset of non-specific symptoms and lack of effective screening, approximately 80% of cases are diagnosed at an advanced stage, defined by the spread of the disease outside the pelvis (International Federation of Obstetrics and Gynecology (FIGO) stage III and IV) [1,2]. Despite efforts and novel treatment strategies the 5-year overall survival of patients with advanced HGSC remains around 40%. On the other hand, when diagnosed at an early stage (FIGO I and II), the 5-year overall survival is 86% and 71%, respectively [2,3]. Biomarkers for early detection of HGSC are, thus, a research priority.

In recent years, extracellular vesicles (EVs) have gained attention for their potential to serve as biomarkers for a wide variety of diseases. EVs are a heterogeneous group of lipid-bound vesicles secreted by cells into the extracellular environment that can be isolated from all body fluids, including blood, free peritoneal fluid (FPF), and ascites. According to their site of formation, biogenesis, and size, EVs are subdivided into three main groups; exosomes, microvesicles, and apoptotic bodies [4]. Isolated samples generally contain a mixture of all subtypes, so the general term “extracellular vesicles” is used as recommended by the International Society of Extracellular vesicles (ISEV) [5,6]. The release of EVs is upregulated in many pathological states, including cancer. They contain tissue-specific signaling molecules (proteins, RNA, and DNA) and mediate intercellular communication not only in tumor microenvironment but also at distant sites [7]. In ovarian cancer, EVs promote tumor growth, metastasis, immune evasion, and the development of chemoresistance [4,7,8,9,10,11]. The composition of EVs reflects the (patho)physiological state of the cell of origin, which together with all the above stated, makes them a promising source of diagnostic, prognostic, and predictive biomarkers that can be obtained by minimally invasive liquid biopsy.

Most studies in ovarian cancer focused on EV concentration and EV proteomic and RNA profiles, which have shown promise for diagnostic biomarker application in ovarian cancer [3,12,13,14,15,16,17]. However, bulk analysis of the EV molecular profile generally fails to meet clinical requirements for diagnostic biomarkers, and recent studies also emphasize the significance of EV size distribution [18,19,20,21,22]. Size-dependent EV proteomic analyses have shown that each size fraction contains unique proteins with distinct biological functions [19,20]. EV size distribution also reflects intricate processes underlying EV biogenesis, which are still poorly understood, although crucial for the development of EV-based diagnostic biomarkers [23,24].

Size distribution analyses of EVs in local fluid and plasma of ovarian cancer patients compared to patients with benign ovarian pathology and healthy controls could aid in understanding the biological functions of EVs and their biomarker potential. Currently, researchers are limited by the available technology for EV isolation and characterization. Van der Pol et al. compared the most widely used techniques—transmission electron microscopy (TEM), flow cytometry, nanoparticle tracking analysis (NTA), and resistive pulse sensing (RPS)—and observed that each technique gave a different concentration and size distribution for the same vesicle sample, primarily caused by differences in the minimum detectable EV size. The minimum detectable vesicle size was smallest with NTA (70–90 nm) [25].

Gercel-Taylor et al. used nanoparticle tracking analysis (NTA) to determine concentration and size distribution of EVs in the sera of ovarian cancer patients, patients with benign ovarian pathology, and healthy controls. They observed a 4-fold increase in the level of total circulating vesicles in ovarian cancer patients. The size range of EVs was similar between the observed groups (50–300 nm), but patients with benign ovarian pathology and healthy controls possessed a greater percentage of large EVs (200–300 nm) [26]. However, their study group comprised only eight ovarian cancer patients.

To address the challenges and opportunities in this rapidly evolving field of EVs, ISEV has updated its Minimal Information for Studies of Extracellular Vesicles (MISEV) guidelines in 2023, following the previous versions published in 2014 and 2018 as MISEV 2014 and MISEV 2018, respectively. MISEV guidelines were the first to show that, in addition to methods for isolation and characterization of EVs, pre-analytical variables (e.g., sample collection, storage, and processing) are also important for accurate results [5,6,27].

In our study, paired pretreatment ascites and plasma samples from 37 patients with advanced HGSC and paired pretreatment free peritoneal fluid (FPF) and plasma samples from 40 controls with benign ovarian pathology (BOP) were analyzed. FPF and ascites represent local fluid. The aim was to evaluate the potential of EV concentration and size distribution measured by NTA in local fluid and plasma as diagnostic biomarkers for HGSC. We also evaluated the potential of peritoneal washing to substitute for FPF, when it is absent.

We observed a significant difference in EV concentration in local fluid, but not in plasma, between HGSC patients and the control group. We found a significant difference in EV size distribution in local fluid and plasma between HGSC patients and the control group. In local fluid, the mean, mode, D10, and D50 were significantly different. In plasma, the mode, D10, and D50 percentile values were significantly different between HGSC patients and the control group. The results of our study indicate that EV concentration in local fluid and EV size distribution in both local fluid and plasma have the potential to differentiate between HGSC and benign ovarian pathology. We also observed significant correlation of EV concentration and size distribution between FPF and peritoneal washing, indicating peritoneal washing may serve as a substitute for FPF when it is absent. To our knowledge, this is the first study of EV size distribution in paired local fluid and plasma samples on a larger cohort of HGSC patients and controls.

## 2. Materials and Methods

### 2.1. Study Design

We conducted a prospective cohort study involving patients with suspected or confirmed diagnoses of advanced HGSC and patients with allegedly benign ovarian pathology (BOP) as a control group. All primary surgical procedures were performed at the Gynaecological Department of University Medical Centre Ljubljana between October 2016 and August 2023. Data collected included patient age and preoperative CA125 tumor marker levels. On the day of the planned primary surgery, blood samples were taken from fasting individuals. In BOP patients, free peritoneal fluid (FPF) samples were aspirated into a sterile syringe at the beginning of laparoscopic surgery. After aspiration of FPF standardized peritoneal washing was performed [28]. In patients with advanced HGSC, ascites samples were taken at the beginning of the primary surgery. Ascites is the pathological accumulation of FPF, present in most cases of advanced HGSC. FPF, peritoneal washing, and ascites represented local fluid.

If the diagnosis of advanced-stage HGSC or BOP in the control group was not confirmed with histopathological examination, patients were excluded from the study.

All patients provided written informed consent prior to study enrolment, and the research adhered to the principles of the Declaration of Helsinki. Our research was approved by the Republic of Slovenia National Medical Ethics Committee (KME 144/12/14).

### 2.2. Sample Collection

To ensure that blood was free of chylomicrons, samples were collected from fasting individuals. Blood samples were taken on the day of planned primary surgery. A 21-gauge needle was used, and the first milliliters of blood were sent for routine clinical analyses. For EV analyses, the next 1.8 mL of blood were collected in BD vacutainer^®^ Citrate blood collection tubes with 3.2% buffered sodium citrate solution (Becton, Dickinson and Company, Franklin Lakes, NJ, USA). In the study group, 50 mL of ascites were taken with a sterile syringe at the beginning of the operation. In the control group, all available FPF was aspirated into a sterile syringe at the beginning of laparoscopic surgery. Standardized peritoneal washing was then performed [28]. Following this procedure, 20 mL of 0.9% NaCl was applied on the uterus, ovaries, and pelvic peritoneum surfaces and left for 2 min in the pelvic cavity. Afterwards, the total aspirable volume of saline solution was aspirated back into the syringe. After aspiration, samples were transferred into sterile conical tubes. The maximum time interval between collection and preparation of samples was 30 min.

### 2.3. Sample Preparation and Storage

Blood and local fluid (ascites, FPF, and peritoneal washing) samples were processed by centrifugation in two stages for 15 min at 2500× *g* at room temperature to remove cells and debris. After each centrifugation, the supernatant was carefully removed, ensuring it was at least 10 mm above the pellet. This process produced platelet-poor plasma (PPP) samples with less than 10^4^ platelets per μL. The same procedure was applied to ascites, FPF, and peritoneal washing samples, which may also contain platelets. To verify the depletion of platelets and absence of hemolysis in samples, standard clinical laboratory tests were used [29]. Supernatants were then frozen in nitrogen vapor following a modified freezing procedure for critical samples [30] and stored at −80 °C until further purification for nanoparticle tracking analysis (NTA). Samples were allowed only one freeze–thaw cycle.

### 2.4. Sample Purification for Nanoparticle Tracking Analysis in Scatter Mode (S-NTA)

Size exclusion chromatography (SEC; IZON qEV original/70 nm) was utilized to separate EVs from other particles. This method is not only effective in recovering EVs with minimal contaminants but also suitable for clinical sample purification due to its simplicity and speed [31,32,33,34]. Thawed samples (0.5 mL) were loaded into the column, followed by the addition of Dulbecco’s phosphate-buffered saline (DPBS, without Ca^2+^ and Mg^2+^). The initial 3 mL of void volume was discarded, and the subsequent 1.5 mL containing the EVs was collected. The elution of plasma and FPF/ascites proteins occurs more slowly, predominantly from 2.5 mL post-void volume. The optimal recovery size for EVs using this column type is between 70 and 1000 nm, effectively excluding particles smaller than 70 nm and removing most contaminating lipoproteins. However, some larger lipoprotein particles may still be present, necessitating further controls. We evaluated the influence of lipoproteins by measuring their concentration in platelet-poor plasma (PPP) and local fluid samples. Particle-enhanced immunonephelometry (Atellica^®^ NEPH 630 System, Siemens Healthineers, Erlangen, Germany) was employed to quantify Apolipoprotein A1 (ApoA1; HDL) and Apolipoprotein B (ApoB; chylomicrons, VLDL, IDL, and LDL). There was no significant correlation between ApoA1 or ApoB levels and EVs concentration or size distribution in PPP and FPF/ascites samples before purification. After sample purification with SEC, ApoA1 and ApoB levels were below the detection limit in all plasma and local fluid samples. Besides lipoproteins, we also evaluated the influence of contaminating proteins in our samples on our results. Protein content was determined using the Bio-Rad Protein Assay in accordance with the Bradford method. Total protein content before purification did not correlate with EVs characteristics in any sample (all *p* > 0.05). After sample purification with SEC, proteins were below the detection limit in all samples.

### 2.5. Quantification of EV Size and Concentration

EV size and concentration were determined using nanoparticle tracking analysis in scatter mode (S-NTA) with a NanoSight NS300 instrument (488 nm laser) connected to an automated sample assistant (both from Malvern Panalytical, Worcestershire, UK). Samples were diluted to achieve an appropriate concentration for accurate particle tracking, maintaining approximately 20–100 particles in the field of view, as recommended by the manufacturer. Each sample underwent four measurements, each with 80 s acquisitions at 25 °C, and the data were processed using NTA software (version 3.3). Camera levels were adjusted per sample, and settings for all readings included a detection threshold of 5, water viscosity, automatic blur size, and an automatic detection range (10.2–15.7 pix). Key measured parameters included particle concentration (particles/mL of cell culture-conditioned media), mode (nm), mean (nm), and percentile values D10, D50, and D90, which indicate the size below which 10%, 50% or 90% of all particle size are found. Reference nanospheres (100 nm polystyrene, Malvern Pananalytical, #LT3100A) were analyzed on the same day to account for day-to-day instrument variation. Raw data were analyzed using NanoSight NTA 3.3 software (Malvern Panalytical, Worcestershire, UK). The concentration of EVs was adjusted considering the dilution factor during sample preparation, including a factor of 3.4 when samples were passed through the qEV column.

EVs were analyzed following our previously published protocol [22] with modifications necessary for clinical samples. The protocol was tested on cell media from three distinct ovarian cancer cell lines, and the results of NTA were evaluated using fluorescence-triggered flow cytometry (FT-FCM) and transmission electron microscopy. Additionally, large volumes of ascites enabled successful validation of clinical samples. In our current study, EV concentration measured by NTA was also positively correlated to total (calcein-positive) EVs detected by FT-FCM (r = 0.871, *p* < 0.001) [17].

All relevant data from our experiments have been submitted to the EV-TRACK knowledgebase (EV-TRACK ID: EV240145).

### 2.6. Statistical Analysis

Continuous variables were summarized using the mean, median, and interquartile range (IQR: 25−75%). Categorical variables were described using frequencies. Pearson’s and Spearman’s rho correlation coefficients were used to evaluate the relationships between continuous variables. The Mann–Whitney U test was applied to compare differences between two independent groups. Receiver operating characteristics (ROC) curve analysis was performed, and the area under the curve (AUC) was calculated to evaluate the biomarker potential of different EV characteristics. Optimal cut-off values for specific EV characteristics in plasma and FPF were identified using the Youden index in ROC curves. All statistical tests were two-sided, with a significance level set at 0.05. Statistical analyses were conducted using IBM SPSS Statistics, version 29 (IBM Corporation, Armonk, NY, USA).

## 3. Results

### 3.1. Patients’ Characteristics

Diagnoses of patients with advanced HGSC and patients with benign ovarian pathology (BOP) that represented control group are summarized in Table 1.

A comparison of basic patients’ characteristics between the study group and control group is presented in Table 2.

### 3.2. Correlation of EV Characteristics between Free Peritoneal Fluid and Plasma in BOP Patients

Using S-NTA, the concentration and size distribution of EVs in free peritoneal fluid (FPF) and plasma of patients with BOP (control group) was determined. There was no correlation between FPF and plasma in the observed EV characteristics. Results are presented in Table 3.

### 3.3. Correlation of EV Characteristics between FPF and Peritoneal Washing in BOP Patients

To evaluate the potential of peritoneal washing as a substitute for FPF when it is absent, we performed standardized peritoneal washing in 14 BOP patients. We found a significant correlation between FPF and peritoneal washing for all observed parameters, except for modal EV size. Results are presented in Table 4.

### 3.4. Correlation of EV Characteristics between Ascites and Plasma in HGSC Patients

We found significant correlation of the mean and D90 between ascites and plasma from patients with advanced HGSC. Results are presented in Table 5.

### 3.5. Comparison of EV Concentration and Size Distribution in Local Fluid and Plasma between Patients with BOP and Patients with Advanced HGSC

We found a significant difference in EV concentration in local fluid between patients with BOP (FPF) and patients with advanced HGSC (ascites) (Figure 1). In FPF, the median EV concentration was 7.88 × 109 (5.80 × 109–1.03 × 1010) and in ascites it was 1.41 × 1010 (9.03 × 109–2.49 × 1010), *p* < 0.001.

We also found a significant difference in all other observed EV characteristics in the local fluid, except for the D90 value, between BOP and HGSC patients. The mean, median, D10, and D50 values were significantly larger in HGSC patients compared to BOP. On the other hand, the concentration of EVs in plasma did not significantly differ between patients with BOP and patients with advanced HGSC. Size distribution analysis, though, showed a significant difference in modal EV size and D10 and D50 values in plasma, but not in the mean and D90 value. The mode, D10, and D50 values in plasma were significantly larger in BOP patients. Results are presented in Table 6.

### 3.6. Concentration and Size Distribution of EVs in Local Fluid and Plasma as Potential Biomarkers for HGSC

We performed receiver operating characteristics (ROC) curve analysis and determined cut-off values for EV characteristics in local fluid and plasma to discriminate between patients with benign ovarian pathology (BOP) and advanced HGSC. At the cut-off value of 83.9 nm for D10 in local fluid, specificity for discriminating advanced HGSC from BOP was 80% and sensitivity was 83.8% (AUC = 0.872, 95% CI = 0.789–0.955) (Figure 2, Table 7).

In plasma, at the cut-off value of 63.7 nm for modal EV size, specificity for discriminating advanced HGSC from BOP was 62.2%, and sensitivity was 87.2% (AUC = 0.783, 95% CI = 0.678–0.888) (Figure 3, Table 8).

## 4. Discussion

To our knowledge, this is the first study to evaluate the potential of EV concentration and size distribution in paired local fluid and plasma samples to serve as diagnostic biomarkers for advanced HGSC. We observed a significantly higher EV concentration in ascites of patients with advanced HGSC compared to EV concentration in free peritoneal fluid (FPF) of BOP patients. Also, EVs in ascites of patients with advanced HGSC were significantly larger. In plasma, a significant difference in the mode, D10, and D50 value was observed between BOP and HGSC patients. In this case, values were larger in BOP patients. The results of our study indicate that EV concentration and size distribution in local fluid and EV size distribution in plasma have potential to serve as diagnostic biomarkers for HGSC.

HGSC, the most common type of ovarian cancer, is still diagnosed at an advanced stage (FIGO stage III, IV) in the majority of cases. It is the deadliest of gynecological cancers with a 5-year overall survival of only 40%. It usually presents late with unspecific symptoms and so far, there is no effective screening strategy available [1,2]. The most commonly used biomarker for ovarian cancer is CA125. It is a powerful predictor of progression-free survival and overall survival and is used for monitoring response to treatment and detecting disease recurrence in ovarian cancer. However, it has a sensitivity of only 50–62% for early stage (FIGO I, II) of ovarian cancer [35]. Novel biomarkers for early detection of HGSC are, thus, a research priority.

EVs are stable, lipid-bound vesicles that reflect the cell of origin and are available for minimally invasive liquid biopsy. In ovarian cancer, EVs are initially secreted into the local fluid but eventually become systemic and can be isolated from both the local fluid and blood [4,7]. Studies on EV concentration and EV proteomic and RNA profiles indicate that EVs have the potential to serve as a novel source of biomarkers also in ovarian cancer [3,12,13,15]. More recently, several authors also suggested that EV size distribution should be considered [18,19,20,21].

We aimed to evaluate the potential of EV concentration and size distribution in the local fluid and plasma to serve as diagnostic biomarkers for HGSC. First, the correlation between local fluid and plasma in both HGSC and BOP patients was analyzed. There was no correlation of EV concentration between local fluid and plasma in either group, suggesting that EVs in plasma originate from different sources. There was, however, correlation in the mean and D90 value between ascites and plasma of HGSC patients, which indicates that larger EVs from ascites have an impact on EV size distribution in the plasma. In BOP patients, no correlation between FPF and plasma was found. To evaluate the potential of peritoneal washing to substitute for FPF, when it is absent, the correlation between FPF and peritoneal washing was assessed. We found significant correlation of all of the observed parameters, except for modal EV size, between peritoneal washing and FPF. Finally, we compared EV characteristics in the local fluid and plasma of HGSC and BOP patients. A significant difference in EV concentration in local fluid between HGSC and BOP patients was observed. EV concentration was higher in ascites of HGSC patients. Size distribution analysis further showed a significant difference in the mode, mean, and D10 and D50 values, and the difference in D90 value was borderline significant. There was no significant difference in EV concentration in the plasma between HGSC and BOP patients. This result is in contrast with a study by Gercel-Taylor et al., which reported a higher concentration of EVs in the sera of ovarian cancer patients as compared to patients with benign ovarian pathology and healthy controls. However, they only included eight ovarian cancer patients [26]. Our results of EV size distribution analysis in plasma have shown significant difference in the mode, D10, and D50 values between HGSC and BOP patients. EVs were significantly larger in the control group. In the mentioned study by Gercel-Taylor et al. the observed size range of EVs was similar between groups, but patients with benign ovarian pathology and healthy controls possessed a greater percentage of large (200–300 nm) EVs [26]. Also, in a study by Zhang et al., characteristics of EVs derived from normal human ovarian epithelial cells and three epithelial ovarian cancer cell lines were compared using NTA. They observed that the normal cell-derived EVs were significantly larger compared to EVs from malignant cells [36].

To assess the diagnostic biomarker potential of EV characteristics in the local fluid and plasma, ROC curve analysis was performed and cut-off values were determined. We found excellent diagnostic performance for the mode, D10, and D50 values in the local fluid and acceptable diagnostic performance for EV concentration and mean EV size in the local fluid, as well as for the mode and D10 value in the plasma. Altogether, the results of our study show that EV concentration in the local fluid and EV size distribution in both local fluid and plasma can help differentiate between HGSC and benign ovarian pathology and may serve in diagnostic biomarker applications for HGSC in the future. Despite known limitations of NTA for EV analysis, the approach to EV characterization used in our present study was previously supported by FT-FCM and transmission electron microscopy (TEM) [17,22].

The main limitation of our study, which was performed on a relatively large cohort of patients, is that we only included patients with advanced HGSC in the study group. To validate our results, further studies including patients with early (FIGO I, II) HGSC and healthy controls are needed. For future research, it would be interesting to study EV size-dependent molecular profiles and their potential to serve as diagnostic biomarkers for HGSC.

## 5. Conclusions

To our knowledge, this study is the first to investigate the potential of EV characteristics in paired local fluid and plasma samples for diagnostic applications in HGSC. The size distribution of EVs reflects the complex processes involved in their biogenesis, which are not yet fully understood, although crucial for the development of EV-based diagnostic tools. We found a significantly higher concentration of EVs in the local fluid of patients with advanced HGSC compared to those with benign ovarian pathology (BOP). Additionally, EVs in ascites of patients with advanced HGSC were notably larger. There was also a significant difference in the size distribution of EVs in plasma between HGSC and BOP patients, with larger values observed in the BOP group. Moreover, the comparison of the relationship between local fluid and plasma EVs in benign versus malignant patients revealed that, in malignant conditions, the plasma more accurately mirrors the size of EVs found in the local fluid. The results of our study suggest that EV characteristics, especially size distribution, in the local fluid and plasma, could serve as part of a composite diagnostic biomarker for HGSC. Size-dependent molecular profiling (RNA and protein content) of EVs might improve diagnostic accuracy; however, further research is needed to confirm our observations and to assess the relevance of EV size distribution in diagnostic biomarker applications for HGSC.

## Figures and Tables

**Figure 1 diagnostics-14-02235-f001:**
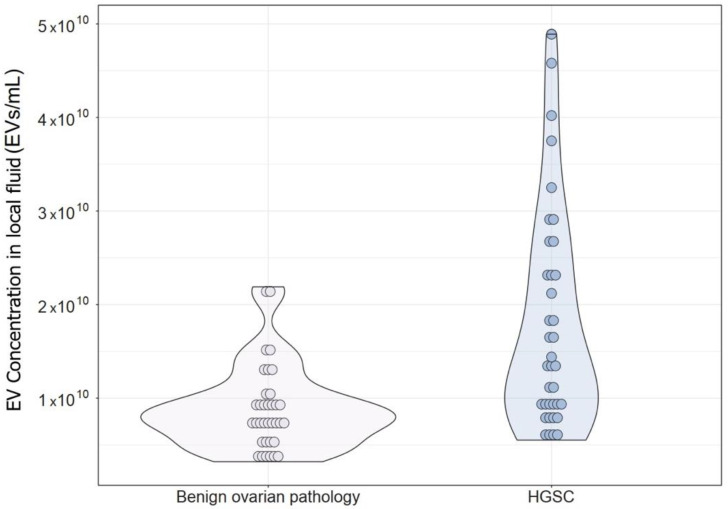
EV concentration in the local fluid of patients with benign ovarian pathology (free peritoneal fluid) and patients with advanced HGSC (ascites).

**Figure 2 diagnostics-14-02235-f002:**
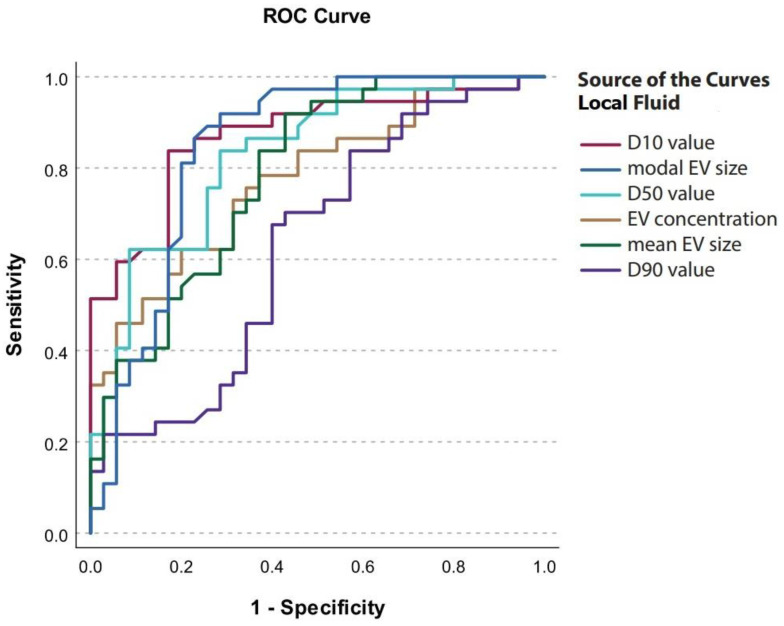
ROC curves for EV characteristics in local fluid.

**Figure 3 diagnostics-14-02235-f003:**
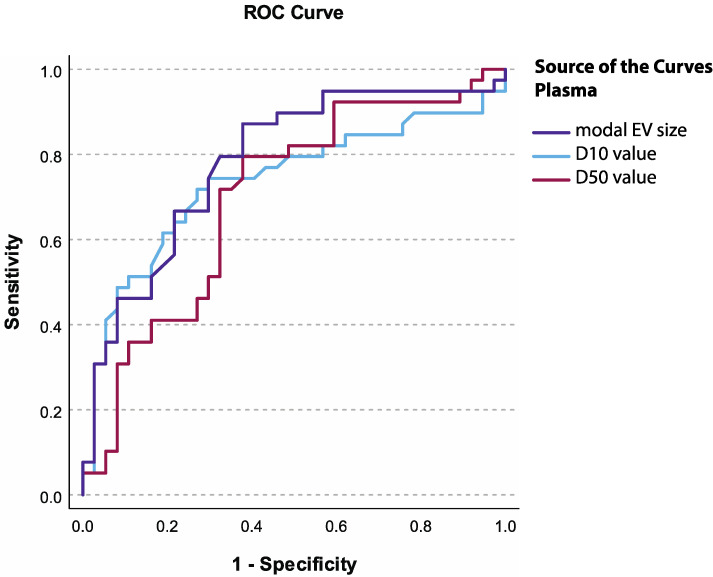
ROC curves for EV characteristics in plasma.

**Table 1 diagnostics-14-02235-t001:** Patients diagnoses.

Study Group		*n*
**HGSC**	**FIGO stage**	**37**
	IIIA, *n* (%) IIIB, *n* (%) IIIC, *n* (%) IVA, *n* (%) IVB, *n* (%)	1 (2.7%) 2 (5.4%) 30 (81%) 1 (2.7%) 3 (8.1%)
	**Ascites**	37 (100%)
**BOP**	**HP**	40
	Endometriotic cyst	9 (22.5%)
	Mucinous cystadenoma	8 (20%)
	Cystic teratoma	7 (17.5%)
	Follicular cyst	5 (12.5%)
	Fibroma	5 (12.5%)
	Paraovarian cyst	2 (5%)
	Corpus luteum cyst	2 (5%)
	Serous cystadenoma	2 (5%)
	**FPF**	35 (87.5%)
	**Peritoneal Washing**	14 (35%)

HGSC—High grade serous carcinoma, BOP—Benign ovarian pathology, FIGO—International Federation of Gynaecology and Obstetrics, IIIA—umor involving one or both ovaries or fallopian tubes or peritoneal cancer with microscopically confirmed peritoneal metastases outside the pelvis and/or metastasis to the retroperitoneal lymph nodes, IIIB—Macroscopic peritoneal metastases that extend beyond the pelvis and that are ≤ 2 cm in largest dimension, with or without positive retroperitoneal lymph nodes, IIIC—Macroscopic peritoneal metastases that extend beyond the pelvis and are > 2 cm in largest dimension, with or without metastasis to retroperitoneal lymph nodes (includes extension of tumor to the capsule of the liver and spleen without parenchymal involvement of either organ), IV—Distant metastases excluding peritoneal metastases, HP—Histopathology, FPF—Free Peritoneal Fluid.

**Table 2 diagnostics-14-02235-t002:** Comparison of baseline patients’ characteristics between study and control group.

Variables		HGSC	BOP	*p*-Value
		Median (25–75%)	Median (25–75%)	
Age	Years	65 (59–74)	35 (29–50)	<0.001 *
Preoperative CA125 level		758 (460–1825)	20 (13–28)	<0.001 *

HGSC—high grade serous carcinoma, BOP—benign ovarian pathology. Comparison of EV characteristics between BOP and HGSC patients was calculated using the Mann–Whitney U test. * Represents < 0.05, which was considered statistically significant.

**Table 3 diagnostics-14-02235-t003:** Correlation of EV characteristics between FPF and plasma of BOP patients.

		BOP Patients (Control Group)		
Variables		FPF	Plasma	*p*-Value
		*n* = 40	*n* = 40	
		Median (25–75%)	Median (25–75%)	
Concentration	EVs/mL	7.88 × 10^9^ (5.80 × 10^9^–1.03 × 10^10^)	3.26 × 10^10^ (2.03 × 10^10^–3.86 × 10^10^)	0.103
Mean	nm	131.2 (109.4–159.9)	89.6 (81.7–96.2)	0.107
Mode	nm	78.4 (68.4–87.7)	69.2 (65.0–76.7)	0.058
D10	nm	74.2 (63.5–81.3)	60.9 (56.1–68.3)	0.406
D50	nm	104.0 (86.1–141.6)	77.4 (71.7–86.2)	0.126
D90	nm	229.9 (180.6–275.1)	131.2 (118.2–146.7)	0.081

BOP—benign ovarian pathology, FPF—free peritoneal fluid, EVs—extracellular vesicles, D10, D50, and D90 are percentile values that represent the size below which 10%, 50%, and 90% of the EV population is found. The correlation of EV characteristics between FPF and plasma was calculated using Pearson’s correlation coefficient.

**Table 4 diagnostics-14-02235-t004:** Correlation of EV characteristics between FPF and peritoneal washing in BOP patients.

		BOP Patients (Control Group)		
Variables		FPF	Peritoneal Washing	*p*-Value
		*n* = 14	*n* = 14	
		Median (25–75%)	Median (25–75%)	
Concentration	EVs/mL	9.11 × 10^9^ (5.05 × 10^9^–1.04 × 10^10^)	1.68 × 10^9^ (1.5 × 10^9^–2.09 × 10^9^)	0.007 *
Mean	nm	129.1 (107.9–151.6)	135.6 (119.4–154.0)	0.009 *
Mode	nm	76.1 (67.7–83.5)	76.0 (70.3–82.3)	0.875
D10	nm	74.4 (64.4–78.0)	72.1 (65.8–81.2)	0.045 *
D50	nm	101.1 (86.8–124.9)	115.3 (87.6–130.4)	0.001 *
D90	nm	206.9 (179.5–248.5)	228.3 (219.1–258.9)	0.001 *

FPF—free peritoneal fluid, EVs—extracellular vesicles, D10, D50, and D90 are percentile values that represent the size below which 10%, 50%, and 90% of the EV population is found. The correlation of EV characteristics between FPF and peritoneal washing was calculated using Pearson’s correlation coefficient. * Represents *p* < 0.05, which was considered statistically significant.

**Table 5 diagnostics-14-02235-t005:** Correlation of EV characteristics between ascites and plasma in HGSC patients.

		HGSC Patients (Study Group)		
Variables		Ascites	Plasma	*p*-Value
		*n* = 37	*n* = 37	
		Median (25–75%)	Median (25–75%)	
Concentration	EVs/mL	1.41 × 10^10^ (9.03 × 10^9^–2.49 × 10^10^)	2.64 × 10^10^ (1.92 × 10^10^–3.65 × 10^10^)	0.163
Mean	nm	166.0 (148.1–184.6)	83.6 (76.9–91.5)	0.015 *
Mode	nm	101.6 (93.2–128.0)	61.7 (59.2–66.3)	0.769
D10	nm	95.9 (84.6–103.9)	54.9 (51.4–58.7)	0.079
D50	nm	150.0 (128.5–166.4)	69.0 (65.3–80.0)	0.125
D90	nm	243.8 (214.2–280.4)	141.1 (122.5–156.4)	0.007 *

HGSC—high grade serous carcinoma, EVs—extracellular vesicles, D10, D50, and D90 are percentile values that represent the size below which 10%, 50%, and 90% of the EV population is found. The correlation of EV characteristics between ascites and plasma was calculated using Pearson’s correlation coefficient. * Represents *p* < 0.05, which was considered statistically significant.

**Table 6 diagnostics-14-02235-t006:** Comparison of EV characteristics in local fluid and plasma between BOP and HGSC patients measured by NTA.

EV Characteristics Measured by NTA	BOP	HGSC	*p*
**Local fluid (FPF/ascites)**	Median (25–75%)	Median (25–75%)	
Concentration (EV/mL)	7.88 × 10^9^ (5.80 × 10^9^–1.03 × 10^10^)	1.41 × 10^10^ (9.03 × 10^9^–2.49 × 10^10^)	<0.001 *
Mean EV size (nm)	131.2 (109.4–159.9)	166.0 (148.1–184.6)	<0.001 *
Modal EV size (nm)	78.4 (68.4–87.7)	101.6 (93.2–128.0)	<0.001 *
D10 value (nm)	74.2 (63.5–81.3)	95.9 (84.6–103.9)	<0.001 *
D50 value (nm)	104.0 (86.1–141.6)	150.0 (128.5–166.4)	<0.001 *
D90 value (nm)	229.9 (180.6–275.1)	243.8 (214.2–280.4)	0.058
**Plasma**			
Concentration (particles/mL)	3.26 × 10^10^ (2.03 × 10^10^–3.86 × 10^10^)	2.64 × 10^10^ (1.92 × 10^10^–3.65 × 10^10^)	0.289
Mean EV size (nm)	89.6 (81.7–96.2)	83.6 (76.9–91.5)	0.085
Modal EV size (nm)	69.2 (65.0–76.7)	61.7 (59.2–66.3)	<0.001 *
D10 value (nm)	60.9 (56.1–68.3)	54.9 (51.4–58.7)	<0.001 *
D50 value (nm)	77.4 (71.7–86.2)	54.9 (51.4–58.7)	0.003 *
D90 value (nm)	131.2 (118.2–146.7)	141.1 (122.5–156.4)	0.167

EVs—extracellular vesicles, NTA—nanoparticle tracking analysis, BOP—benign ovarian pathology, HGSC—high grade serous carcinoma, FPF—free peritoneal fluid, D10, D50, and D90 are percentile values that represent the size below which 10%, 50%, and 90% of the EV population is found. The comparison of EV characteristics between BOP and HGSC patients was calculated using the Mann–Whitney U test. * Represents *p* < 0.05, which was considered statistically significant.

**Table 7 diagnostics-14-02235-t007:** Cut-off values, corresponding specificity and sensitivity and AUC for EV characteristics in local fluid to discriminate between BOP and advanced HGSC.

EV Characteristic in Local Fluid	Cut-Off Value	Sensitivity	Specificity	AUC	95% CI
Concentration (EVs/mL)	1.07 × 10^10^	0.622	0.800	0.777	0.672–0.882
Mean (nm)	133.3	0.919	0.571	0.786	0.682–0.891
Mode (nm)	87.8	0.865	0.771	0.844	0.748–0.941
D10 (nm)	83.9	0.838	0.800	0.872	0.789–0.955
D50 (nm)	122.7	0.838	0.571	0.830	0.736–0.923
D90 (nm)	234.0	0.676	0.600	0.630	0.499–0.760

EV—extracellular vesicle, AUC—area under the curve, CI—confidence interval. D10, D50, and D90 are percentile values that represent the size below which 10%, 50%, and 90% of the EV population is found. Optimal cut-off values for specific EV characteristics were identified using the Youden index in ROC curves.

**Table 8 diagnostics-14-02235-t008:** The cut-off values, corresponding specificity and sensitivity, and the AUC for EV characteristics in plasma to discriminate between BOP and advanced HGSC.

EV Characteristic in Plasma	Cut-Off Value	Sensitivity	Specificity	AUC	95% CI
Mode (nm)	63.7	0.872	0.622	0.783	0.678–0.888
D10 (nm)	57.6	0.718	0.676	0.736	0.619–0.852
D50 (nm)	71.5	0.795	0.622	0.698	0.578–0.818

EV—extracellular vesicle, AUC—area under the curve, CI—confidence interval. D10, D50 and D90 are percentile values that represent the size below which 10%, 50% and 90% of EV population is found. Optimal cut-off values for specific EV characteristics were identified using the Youden index in ROC curves.

## Data Availability

The original contributions presented in the study are included in the article, further inquiries can be directed to the corresponding author. We have submitted all relevant data of our experiments to the EV-TRACK knowledgebase (EV-TRACK ID: EV240145).

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
