# Peer review of "Extracellular Vesicle Characteristics in Local Fluid and Plasma Measured by Nanoparticle Tracking Analysis Can Help Differentiate High-Grade Serous Carcinoma from Benign Ovarian Pathology"

_diagnostics, 2024, doi:10.3390/diagnostics14192235_

Round 1

Reviewer 1 Report

Comments and Suggestions for Authors

Based on the measurement by track analysis of vesicle size and concentration, the authors draw far-reaching conclusions about the use of these characteristics for the diagnosis of ovarian cancer. This kind of work was popular in the 2000s, when, based on the concentration of extracellular DNA, some self-confident authors proposed to diagnose malignant neoplasms in a similar way. However, over the past quarter century, it has become clear that these nonspecific indicators cannot be used in clinical practice because they produce high values of both false-positive and false-negative results.

The future of liquid biopsy lies in the search for specific markers in the composition of vesicles - proteins and nucleic acids, about which more than a thousand reviews are already reflected in the Pubmed database. I recommend rejection of the manuscript.

Reviewer 2 Report

Comments and Suggestions for Authors

It is a known phenomenon that cancer cells produce extracellular vesicles of different size distribution and cargo compared to normal cells. However, the study has been well prepared, performed and presented in the paper. Of course the concentration and size distribution of EVs in peritoneal fluid/ascites/washing fluid has no value in diagnosis and differentiation of adnexal masses. But in the plasma it could be a valuable tool. It would be much more interesting if Authors  performed the validation of the diagnostic accuracy of plasma EVs  for BOT and early (FIGO I/II) patients, as they indicated in the limitations of their study. The only doubt is how did Authors avoid contamination of the fluid samples with the blood.

Round 2

Reviewer 1 Report

Comments and Suggestions for Authors

This manuscript cannot be improved by a small revision because it is based on an erroneous hypothesis: you cannot diagnose cancer by the size of vesicles!